# Systematic design and experimental demonstration of bianisotropic metasurfaces for scattering-free manipulation of acoustic wavefronts

Junfei Li[1], Chen Shen [1], Ana Díaz-Rubio[2], Sergei A. Tretyakov[2] & Steven A. Cummer[1]

Recent advances in gradient metasurfaces have shown that by locally controlling the bianisotropic response of the cells one can ensure full control of refraction, that is, arbitrarily redirect the waves without scattering into unwanted directions. In this work, we propose and experimentally verify the use of an acoustic cell architecture that provides enough degrees of freedom to fully control the bianisotropic response and minimizes the losses. The versatility of the approach is shown through the design of three refractive metasurfaces capable of redirecting a normally incident plane wave to 60°, 70°, and 80° on transmission. The efficiency of the bianisotropic designs is over 90%, much higher than the corresponding generalized Snell's law based designs (81%, 58%, and 35%). The proposed strategy opens a new way of designing practical and highly efficient bianisotropic metasurfaces for different functionalities, enabling nearly ideal control over the energy flow through thin metasurfaces.

[1] Department of Electrical and Computer Engineering, Duke University, Durham, North Carolina 27708, USA. [2] Department of Electronics and Nanoengineering, Aalto University, P. O. Box 15500, FI-00076 Aalto, Finland. These authors contributed equally: Junfei Li, Chen Shen. Correspondence and requests for materials should be addressed to S.A.T. (email: sergei.tretyakov@aalto.fi) or to S.A.C. (email: cummer@ee.duke.edu)

The ability to fully control the behavior of classical waves (e.g., electromagnetic and acoustic waves) has long been desired and is at present a highly active research area. Among numerous routes, metamaterials have served as a primary approach in recent years[1,2]. The possibilities are enabled by engineering subwavelength structures with local resonance to achieve arbitrary effective parameters not found in nature. In contrast to the volumetric modulation using metamaterials, two-dimensional arrangements of subwavelength cells offer an alternative solution of molding wave propagation within a planar or nearly flat geometry. These two-dimensional patterned surfaces, termed metasurfaces, have facilitated unprecedented possibilities for controlling waves at will[3,4]. One of the most attractive aspects of metasurfaces is the ability to engineer the scattered wavefronts by packing phase shifts along the gradient metasurface, which have awakened interest as an approach for the design of lenses, beam splitters, and more[5,6].

In both electromagnetic[7–9] and acoustic[10–16] metamaterials, the conventional gradient metasurface design approach is based on the implementation of local phase modulation which dictates the behavior of outgoing waves according to the generalized Snell's law (GSL)[12]. In acoustics, various unit cell topologies have been proposed to achieve a homogenized effective index to control the local transmitted or reflected phase[10–12,14–19]. They have been applied to acoustic devices for different functionalities, such as wavefront manipulation[10–16], sound absorption[16,19,20], asymmetric transmission[21], and cloaking[22,23]. However, the efficiency of phase-shift devices is fundamentally restricted by the reflection and scattering into unwanted directions. To enable better performance, many approaches have been applied to improve the transmission of the unit cells through impedance matching[17,24–28].

However, recent work has shown that the local phase gradient alone cannot provide full control over the scattered wave[29–35].

Consider anomalous refraction as an example, which is the simplest functionality offered by gradient metasurfaces in transmission. For an optimal performance, the metasurface must transmit all the illuminating energy into another arbitrary direction. As was pointed out for electromagnetic and acoustic waves, the fundamental limitation associated with all conventional GSL designs originates in the impedance mismatch between incident and refracted waves. To overcome the problem, one has to control not only the phase gradient along the metasurfaces but also the impedance matching between the incident and the desired scattered waves.

Rigorous analysis of the problem has shown that the macroscopic impedance matching required for theoretically perfect anomalous refraction of plane waves can be realized if the metasurface exhibits bianisotropy: magneto-electric coupling for electromagnetic metasurfaces[30–32] and Willis coupling for the acoustic counterpart[29,36,37]. The bianisotropic response can be implemented by asymmetric unit cells, where the scattered fields are different depending on the direction of illumination. For electromagnetic metasurfaces, typical solutions are based on three cascaded impedance layers. By independently controlling the impedance of each layer, the asymmetric response can be fully controlled[38,39]. These structures have been numerically and experimentally verified. In acoustics, however, practical design or experimental realization of perfect anomalous refractive metasurfaces has remained scarce.

Interest in bianisotropy in acoustics begun recently[37,40,41]. Bianisotropy provide two new possibilities for acoustic metasurfaces: independently control the reflection and transmission phases[40], or the difference in the reflection phases[41]. A deep analysis of the physics behind this phenomenon and clear analogy between electromagnetic and acoustic bianisotropy has been reported[37]. These results indicate that acoustic bianisotropy could bring new directions for designing efficient metasurfaces, as in the electromagnetic counterpart.

The next step is designing acoustic metasurfaces, which benefit from bianisotropy. Bianisotropic meta-atoms in macroscopic acoustic metasurfaces for wavefront modulation have been recently proposed by Koo et al.[40] where different gradients were applied in reflection and transmission to control the reflected and transmitted wavefronts simultaneously. However, part of the energy will be scattered without control. Scattering-free manipulation of the wavefronts requires strict control over the meta-atom properties depending on the desired transformation[29]. An approach for perfect anomalous refraction was theoretically proposed using three membranes[29]. However, the surface tension, uniformity, and durability, etc. of the membranes are extremely difficult to control and it is questionable whether this design can be practically realized.

To design bianisotropic metasurfaces, one has to deal with three important issues. First, the tangential dimension of the meta-atom must be deeply subwavelength for ensuring a smooth gradient profile in most cases. Second, meta-atoms must ensure complete control of the scattered waves. Recent electromagnetic and acoustic studies[29,30,32,38,39] have shown that full control of the bianisotropy requires at least three degrees of freedom in the particle design. Finally, the intrinsic losses associated with the resonant elements can affect the overall efficiency. Therefore, although the three-membrane topology satisfies the minimum requirements for obtaining arbitrary bianisotropic response, the structures become resonant, inducing a great amount of losses inside the structures.

In view of the current problems and previously reported designs, here we propose a versatile platform for bianisotropic metasurfaces based on four independent resonators. This topology avoids operation near resonance frequencies and consequently allows smaller elements (in the tangential dimension) with lower losses. The resonators can be easily controlled by changing the geometrical dimensions. The validity of the proposed bianisotropic particle is tested with the design of three refractive metasurfaces that fully redirect a normally incident plane wave into 60°, 70°, and 80° on transmission. The power efficiency of the bianisotropic designs is over 90%, much higher than the corresponding conventional GSL-based designs (81%, 58%, and 35%). The design is valid even with coarse sampling of the required impedance profile and is robust against fabrication errors. We also experimentally characterize a bianisotropic metasurface for scattering-free acoustic anomalous refraction, where 97% of the transmitted energy goes to the desired direction and < 2% of the energy is reflected.

## Results

**Limitations of the GSL.** An ideal metasurface can refract incident energy into an angle $\theta_t$ with 100% efficiency, as shown in Fig. 1a. It requires macroscopic impedance matching between the incident wave and refracted waves, which cannot be fulfilled by a phase shift only, even with unitary transmission from individual cells. Therefore, GSL-based metasurfaces, which work with high efficiency at low incidence angles, have poor efficiency at larger angles. Figure 1b shows the theoretical efficiency limitation for GSL-based metasurfaces illuminated normally as a function of the angle of refraction. This limitation is inherent to the design approach and does not depend on the cell topology. This limitation can be solved, however, by introducing bianisotropy into the building blocks of the metasurface.

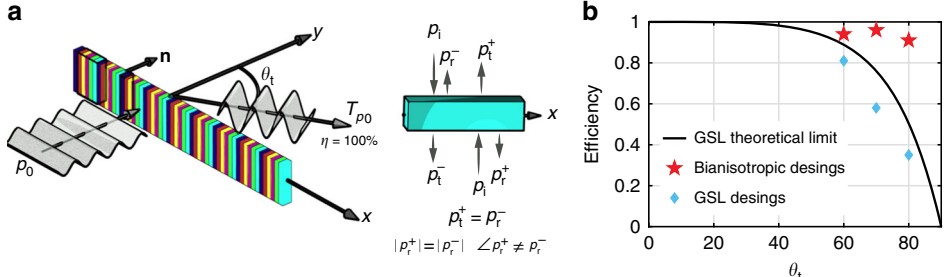

**Fig. 1** Desired metasurface and achieved performance. **a** Illustration of the desired performance of a perfect anomalous refractive metasurface. $\eta$ denotes the energy efficiency. All the energy is pointing out toward the desired direction with no parasitic scattering. The inset shows the bianisotropic response of the elements of the metasurface, that is, the asymmetric response for incident waves from opposite directions (the phase of the reflected wave are different). **b** Comparison of the efficiency for anomalous transmission metasurfaces. Bianisotropic designs show great advancement especially for large deflection angles. Realized efficiencys are slightly lower than the theoretical limit as a result of discretization

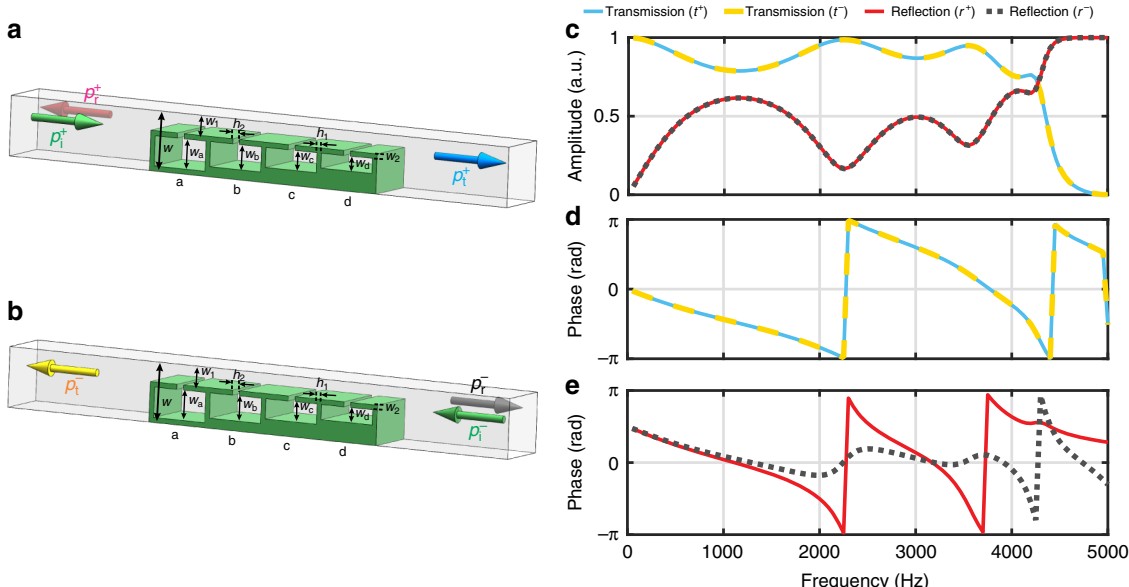

**Fig. 2** Study of a bianisotropic acoustic cell. **a** Geometry of a cell with four side-loaded resonator. The height of the Helmholtz resonators is varied to create different bianisotropic responses. Definition of the forward (+) and backward (−) illuminations. **b** Amplitude and phase of the transmission and reflection coefficients of an arbitrary cell. The dimensions of the cell are: $w = 12$ mm, $h_2 = 1.5$ mm, $w_2 = 1$ mm, $h_1 = 1$ mm, $w_1 = 4$ mm, $w_a = 6$ mm, $w_b = 5$ mm, $w_c = 4$ mm, and $w_d = 3$ mm

**Non-resonant bianisotropic acoustic cells**. The cell architecture that we use to ensure asymmetry, shown in Fig. 2a,b, is based on a straight channel with side-loaded resonators. Similar structures with identical resonators have been used to slow down the speed of sound in the channel for output phase control[18,19,42], or absorption enhancement[43,44]. However, they do not exhibit bianisotropy, as the metasurfaces have identical response from opposite directions. In the most general case, the relation between the fields at both sides of a lossless array of bianisotropic cells placed at $y = 0$ can be expressed as

$$\begin{bmatrix} p^+(x,0) \\ p^-(x,0) \end{bmatrix} = \begin{bmatrix} Z_{11} & Z_{12} \\ Z_{21} & Z_{22} \end{bmatrix} \begin{bmatrix} \mathbf{n} \cdot \mathbf{v}^+(x,0) \\ -\mathbf{n} \cdot \mathbf{v}^-(x,0) \end{bmatrix} \quad (1)$$

where $\mathbf{n}$ is the normal vector of the metasurface, $Z_{ij}$ are the components of the impedance matrix, and the $\pm$ sign refers to the fields at both sides of the metasurface. It is noteworthy that for such a linear time-invariant system under study, reciprocity requires $Z_{12} = Z_{21}$ and we assume this condition throughout. The cell will have bianisotropic response if $Z_{11} \neq Z_{22}$ and this

condition can be satisfied if the acoustic cells has structural asymmetry [see Fig. 2a].

From the analysis of the bianisotropic requirements dictated by the impedance matrix, we can see that with the proposed topology three resonators is the minimum requirement which allows to implement any desired response (Supplementary Note 2). However, to obtain extreme asymmetric response required by some gradient metasurfaces, the resonators have to work near their resonant frequencies and this makes it difficult to control their responses and increase loss. The required resonance also puts constraints on the physical dimensions and cause robustness issues to the practical designs. In order to mitigate these practical limitations, we propose a four side-loaded resonators particle, as shown in Fig. 2a. In this structure, the width and height of the neck, $h_2$ and $w_2$, are fixed in the four resonators; the width of the cavities $h_3$ is also fixed; the height of the air channel $w_1$ and the height of the resonators $w_a$, $w_b$, $w_c$, and $w_d$ can be varied to control the asymmetry; and the wall thickness of the unit cell $h_1$ is fixed and will be defined with the fabrication limitations. All the thicknesses are less than half of a wavelength.

A simple way to study the bianisotropic response of the proposed particle is by analyzing the scattering produced by the particle. The scattering of the particle can be expressed in terms of the scattering matrix as

$$\begin{bmatrix} p_s^+ \\ p_s^- \end{bmatrix} = \begin{bmatrix} r^+ & t^- \\ t^+ & r^- \end{bmatrix} \begin{bmatrix} p_i^+ \\ p_i^- \end{bmatrix} \qquad (2)$$

where $p_i^\pm$ represent the amplitude of the forward and backward incident plane waves, $p_s^\pm$ is the amplitude of the scattered fields at both sides of the particle (i.e., $p_s^+ = p_r^+ + p_t^-$ and $p_s^- = p_r^- + p_t^+$), $t^\pm$ represent the local transmission coefficients, $r^\pm$ are the reflection coefficients (the relation between the scattering matrix and the impedance matrix is detailed in Methods). Figure 2c–e shows the transmission and reflection amplitudes and phases for a particle defined by $h_2 = 1.5$ mm, $w_2 = 1$ mm, $h_1 = 1$ mm, $w_1 = 4$ mm, $w_a = 6$ mm, $w_b = 5$ mm, $w_c = 4$ mm, and $w_d = 3$ mm. For lossless and reciprocal particles, the transmission coefficients and reflection coefficients satisfy $t^+ = t^- = t$, $|t|^2 + |r^\pm|^2 = 1$, and $r^+ t^* + t r^{-*} = 0$ (see Supplementary Note 1). The analysis of Fig. 2c–e shows that only the phase of the reflection is different for opposite directions and this reflection phase asymmetry is a clear signature of bianisotropy[37,45]. Although there is also asymmetry in the orthogonal direction of the unit cells, it can be ignored as long as the width of the channel is significantly smaller than a wavelength. Also, as there are walls between adjacent cells, the wave does not propagate along the orthogonal direction inside the metasurface. Therefore, all the cells in the bianisotropic metasurfaces can be designed individually.

**Design of acoustic bianisotropic metasurfaces**. To demonstrate the applicability of the proposed bianisotropic cell, in what follows, we will design refractive metasurfaces for steering a normal incident wave ($\theta_i = 0°$) into a transmitted wave propagating at $\theta_t$. For a perfect refractive metasurface (with energy efficiency

$\eta = 100\%$), all the incident energy is redirected to the desired direction. This condition, equivalent to energy conservation in the normal direction, requires the macroscopic pressure transmission coefficient to satisfy $T = 1/\sqrt{\cos\theta_t}$. Detailed definition of the macroscopic transmission coefficient can be found in the Method section. Imposing the boundary conditions dictated by Eq. (1), we can calculate the value of the impedance matrix at each point of the metasurface as

$$Z_{11} = iZ_0 \cot(\Phi_x x) \qquad (3)$$

$$Z_{12} = i \frac{Z_0}{\sqrt{\cos\theta_t}} \frac{1}{\sin(\Phi_x x)} \qquad (4)$$

$$Z_{22} = i \frac{Z_0}{\cos\theta_t} \cot(\Phi_x x) \qquad (5)$$

where $\Phi_x = k \sin\theta_t$ is the phase gradient along the metasurface and $Z_0$ is the characteristic acoustic impedance of the background medium. The period of the metasurface can be calculated as $D = 2\pi/\Phi_x$. Equation (5) shows that $Z_{11}$ is not equal to $Z_{22}$, so the bianisotropic response is required.

The local transmission and reflection coefficients, which have to be implemented by the bianisotropic unit cell can be calculated as

$$t = \frac{2\sqrt{\cos\theta_t}}{1 + \cos\theta_t} e^{i\Phi_x x}, \qquad r^+ = \frac{1 - \cos\theta_t}{1 + \cos\theta_t} e^{-i2\Phi_x x}, \qquad (6)$$

and

$$r^- = \frac{1 - \cos\theta_t}{1 + \cos\theta_t}. \qquad (7)$$

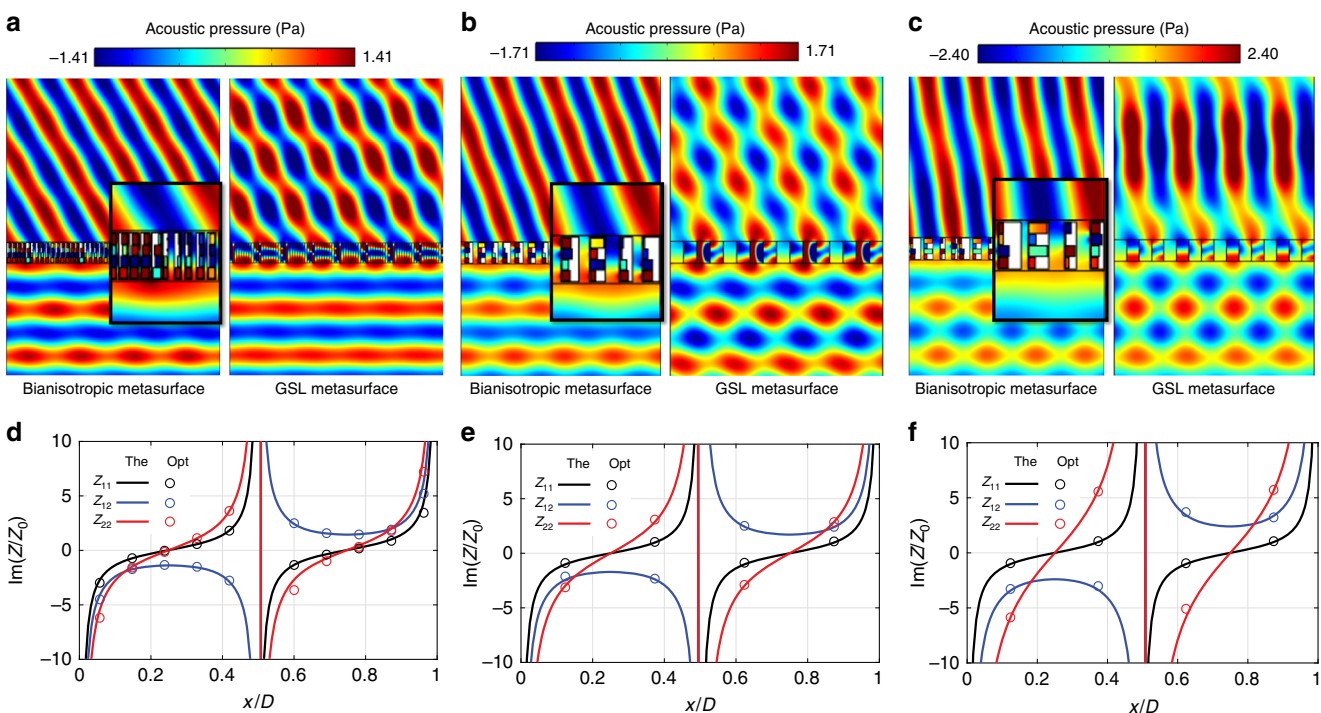

**Fig. 3** Bianisotropic metasurfaces for scattering-free anomalous refraction. **a–c** represent the numerical simulation of the total pressure field for bianisotropic metasurfaces and GSL metasurfaces when $\theta_t = 60°$, $70°$, and $80°$. The insets show the phase evolution inside the metasurface. **d–f** represent the impedance matrices profile for $\theta_i = 0°$ and $\theta_t = 60°$, $70°$, and $80°$. Here The denotes theoretical requirements, whereas Opt denotes the impedance values achieved by structure optimization

These expressions demonstrate the strict relation, not only in phase but also in magnitude, of the local reflection and transmission coefficients with the desired angle of refraction. Moreover, one can see the difference in phase between the reflection coefficients in forward and backward directions, which appear as a consequence of the bianisotropy[41]. It is important to notice that gradient metasurface described by Eq. (6) is different from conventional designs based on the generalize refraction law[7,8,16] and the gradient bianisotropic metasurface in ref. [40]. This asymmetric behavior can be achieved by using bianisotropic cells proposed in this work. It is worth noting here that the reflection coefficient has twice the tangential wave number of the transmission coefficient, indicating that energy exchange along the metasurface is carried out by surface waves on the reflection side. This is related to the idea of controlling reflection by launching an auxiliary surface wave to achieve scattering-free wave manipulation[32].

The first design presented in this work corresponds to $\theta_t = 60°$. In this case, the required values for the components of the impedance matrix are represented in Fig. 3d. The operating frequency is chosen to be 3000 Hz that makes the period of the metasurface $D = 13.2$ cm ($D/\lambda = 1.15$). We use 11 cells along the period for implementing the spatial dependent bianisotropic response, so the width of the unit cell is $w = D/11 = 12$ mm ($w/\lambda = 0.10$). In the discretization process, we choose the cells to have the impedance values at $x_n = (n − 0.375)w$, where $n$ denotes the index of the cell, to avoid points where the ideal impedance matrix diverges. For the design of the physical dimensions, genetic algorithm optimization is used to define $w_1$, $w_a$, $w_b$, $w_c$, and $w_d$ so that the calculated impedance matrix matches the theoretical requirements. The physical dimensions of the final design and their corresponding transmission/reflection coefficients are summarized in the Supplementary Material. More details of the genetic algorithm optimization process, such as stop criteria, repeatability and evolution of the cost function can be found in the Methods.

From Fig. 3d we can see that the required impedance matrix of the perfect metasurface is closely approximated by our unit cells. It should be noted that the metasurface is discretized and approximated with a finite number of cells, and the performance of the metasurface can be possibly enhanced by using a larger number of cells with better spatial resolution.

Full-wave simulations are performed to verify our design. The real part of the simulated acoustic pressure field for our first structure is represented in Fig. 3a, where nearly total energy transmission is observed. The simulated amplitude ratio $T$ achieved with our real structure is 1.365, as compared with the theoretically ideal value of 1.414, indicating that 93% of the incident energy is transmitted to the desired direction. This value is higher than the theoretical upper limit of 89% power transmission for conventional GSL based designs [see Fig. 1b]. Solid line in Fig. 1b shows the theoretical efficiency for a metasurface illuminated normally as a function of the angle of refraction. With the purpose of comparison, we use a simulation of a discretized impedance-matched design based on the GSL, confirming that in the conventional metasurfaces only 81% of the input energy is transmitted in the desired direction, with the remainder going into reflection and other diffractive modes. Figure 3a shows the comparison between the responses of both designs, where we can clearly see the improvement obtained with the bianisotropic design.

Despite the piecewise constant and approximate realization of the theoretically ideal impedance profile, this practical structure nearly realizes perfect, lossless transmission of energy in the desired direction. This shows that realistic structures can perform significantly better than conventional metasurfaces. Critically, it

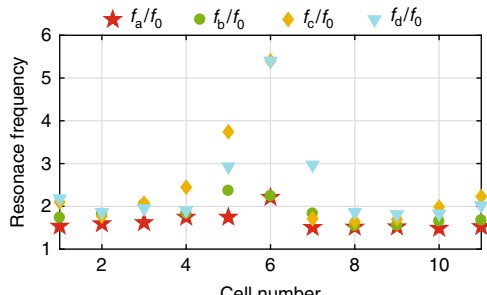

**Fig. 4** Normalized resonance frequency of the individual resonators of the scattering-free anomalous refractive metasurface designes for $\theta_i = 0°$ and $\theta_t = 60°$. All the resonators are working out of the resonant frequency to avoid high losses

also shows that good performance of a wavefront transformation metasurface does not require perfect realization of the ideal impedance profile. A close and piecewise approximation will suffice in our design.

Figure 4 depicts the normalized resonance frequency of each individual resonators with respect to the operation frequency $f_0$ (3000 Hz). It is important to notice that none of the resonators is working near the resonance; thus, the design will be less sensitive to the losses than other resonant designs, e.g., the three-membrane proposal[29]. The performance of the design is also confirmed in simulation by considering viscous loss, as it is the inherent loss of the structure, which is inevitable in the experiments (Supplementary Note 4). In addition, due to the high resonance frequencies of the resonators, their size allows smaller width of the cells, i.e., it is easier to implement gradient metasurfaces with this topology.

To better show the large efficiency enhancement of the bianisotropic metasurface over conventional GSL-based designs, we designed another two cases with larger deflection angles, where the metasurfaces steer the incident beam to $\theta_t = 70°$ and $\theta_t = 80°$, respectively. For these two cases, the metasurfaces are sampled coarsely with only four cells within one period. The theoretical requirements (lines) and the achieved values (dots) of the impedance matrices for both cases are shown in Fig. 3e,f. Detailed dimensions and relative errors can be found in Supplementary Tables 2 and 3. Figure 3b, c show the simulated results of the bianisotropic designs and the corresponding GSL-based designs with ideal impedance matched cells and the same resolution. Energy efficiencies of the bianisotropic designs achieved 96% and 91% for 70° and 80° cases, whereas the corresponding numbers for GSL designs are 58% and 35%, respectively. The imperfect scattered field is caused by the reflection from the metasurface, which is due to non-ideal implementation of the metasurface. We note that, however, the power flow normal to the surface (the conserved quantity that defines energy efficiency) in these reflections is low and contributes little to the overall energy efficiency of the metasurface. As the deflection angles, i.e., 70° and 80°, are large, the reflection amplitudes of 0.35 and 0.6, respectively, their contributions to normally directed power follow is only 4% and 6%. In other words, the high efficiency is still maintained even though the reflected field amplitudes are not negligible. It is noteworthy that GSL-based designs are carried out by impedance matched cells with precise phase control and the efficiency values are expected to be even lower for real structures. We can see that even with such a coarse representation of the impedance profile and non-negligible relative error, the bianisotropic designs achieved much higher efficiency than the conventional ones.

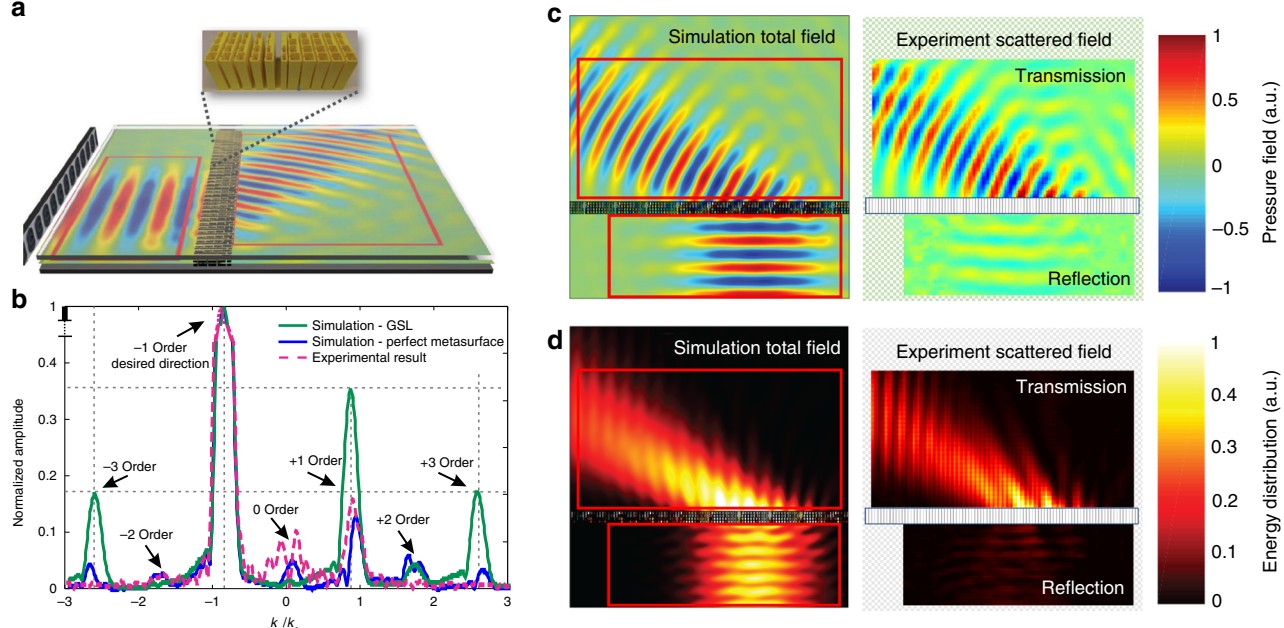

**Fig. 5** Experimental setup and results. **a** Schematic representation of the experimental setup and a period of the fabricated sample. **b** Comparison between the normalized scattering of the bianisotropic metasurface (experimental and numerical) and a GSL design. Analysis of the real part **c** and magnitude square **d** of the experimental pressure field and the comparison with the numerical simulations

This offers huge advantage for practical realizations, especially in the high frequency or ultrasound range where fabrication capabilities are limited.

**Experimental measurements**. Measurements were carried out to characterize the design experimentally and confirm its scattering-free property. As an example, we picked the 60° case. The experimental setup and one period of the fabricated sample is shown in the Fig. 5a. The measured transmitted pressure field Fig. 5c and energy distribution Fig. 5d are compared with the corresponding simulated fields. Good agreement between simulation and experiment is observed and the small discrepancies can be attributed to fabrication errors and inevitable losses in the lab environment. The experimental results show that unwanted diffraction orders are greatly suppressed and all the transmitted energy is concentrated in one direction. To confirm that our metasurface is reflection-free, the reflected field is also measured. The reflection caused by the metasurface is obtained by scanning the reflected region in the empty waveguide and the field with the metasurface, and then calculating the difference between the two measured fields. The result shows that only 2% of the energy is reflected.

To demonstrate the performance of the metasurface, the normalized energy distribution on each direction is further calculated by performing Fourier transform along the line right behind the metasurface and the result is shown in Fig. 5b. The experimental result shows an excellent consistency with simulations, with most of the energy localized in the desired direction (− 1 order) and other diffraction modes are severely suppressed, including the high-order evanescent modes, which are visible, because we are processing the fields very close to the surface.

The normalized energy distribution of a GSL-based metasurface using impedance-matched lossless effective medium computed from the same simulation is also shown in Fig. 5b for comparison, where undesired diffraction orders can be clearly observed. It should be noted that this number is calculated based on unit cells characterized by matched impedance and ideal

refractive indices, and gives the performance limit of conventional designs. The bianisotropic metasurface proposed here therefore provides an alternative route of overcoming the power efficiency limitation and reduce the parasitic energy spread into undesired directions.

## Discussion

In summary, we design and experimentally demonstrate an acoustic metasurface cell that provides full control of the bianisotropic response and minimizes the implementation losses by ensuring that the individual resonators work below the resonant frequency. The response of the cells, controlled by the physical sides of the four side-loaded resonators and the width of the channel, can be adjusted to provide any scattering requirement. The tangential dimension for the cell is deep subwavelength (1.2 cm for 3000 Hz, $\lambda/10$) compared with previously proposed meta-atoms in ref. [40] (7 cm for 1300 Hz, $\lambda/4$), such that it is readily to be applied to the cases where more complicated impedance profiles are needed. For a specific asymmetric response, a carefully implemented genetic algorithm optimization method calculates the physical dimensions of the unit cell.

In addition, we have demonstrated the first design and realization of bianisotropic acoustic metasurfaces for scattering-free wavefront manipulations. Three perfect metasurfaces for wavefront modulation (with deflection angles of 60°, 70°, and 80°) are designed based on the theory. The performance is validated with numerical simulations, showing great advancement in energy efficiency (93%, 96%, and 91%) over conventional GSL-based designs (89%, 58%, and 35%), especially at large deflection angles. The scattering-free property of the bianisotropic metasurface is further verified experimentally. The designed metasurface is shown to be able to steer all the energy to the desired direction with almost no reflection or unwanted scattering.

It is interesting to note that there the asymmetric matching conditions required for perfect anomalous transmission can also be fulfilled with the use of parity-time (PT)-symmetric structures[46]. In electromagnetics, this possibility was previously

noted[31,47]. However, realizations of this PT approach requires the use of active elements. We also want to emphasize that the proposed design scheme is not restricted to wave steering and can be readily extended to other applications. For example, similar so-called perfect metasurfaces can be designed to achieve sound focusing without scattering, acoustic skin cloaking with low-energy dissipation, and arbitrary acoustic field generation with high energy efficiency, among many others. In general, by considering non-local near field coupling and allowing the most general form of the cell's impedance matrix, it will be possible to overcome the efficiency drawbacks in the existing metasurface designs. Furthermore, as our bianisotropic design approach performs well even with a very coarse approximation of the continuous impedance profile, it offers great advantage in the ease of fabrication, especially for applications requiring a complicated field distribution, or extension to high frequency ranges. We would also like to point out that the proposed structure is not unique for realizing bianisotropic impedances, therefore improved lower loss structures are an important subsequent step toward applications. We believe that the bianisotropic metasurface concepts can largely expand the family of acoustic metasurfaces and open up new sound manipulation capabilities based on the versatile platform that can offer.

## Methods

**Transfer matrix of the bianisotropic unit cell.** An analytic expression of the transfer function of the unit cells is developed to facilitate the design of the wavefront transformation metasurface. The geometry of a unit cell is shown in Fig. 2a,b, where $h_1$ is the thickness of the shell, $h_2$ is the width of the neck, $h_3$ is the length of the cavity, $w$ is the height of the unit cell, and $w_1$ and $w_2$ are the height of the channel and neck, respectively. The height of each individual Helmholtz resonator, $w_{a,b,c,d}$, can be different as asymmetric geometry of the unit cell is required by the bianisotropic metasurface.

The relationship for the pressure and volume velocity of the incident and transmitted waves can be expressed as:

$$\begin{bmatrix} p^+ \\ \mathbf{n} \cdot \mathbf{u}^+ \end{bmatrix} = \begin{bmatrix} M_{11} & M_{12} \\ M_{21} & M_{22} \end{bmatrix} \begin{bmatrix} p^- \\ \mathbf{n} \cdot \mathbf{u}^- \end{bmatrix} \quad (8)$$

where $\mathbf{u}^\pm = w v^\pm$, and $[M]$ is the total transfer matrix that can be written as:

$$[M] = [M_{\text{in}}][N_0][M_{\text{a}}][N_0][M_{\text{b}}][N_0][M_{\text{c}}][N_0][M_{\text{d}}][N_0][M_{\text{out}}]. \quad (9)$$

Here, $[M_{\text{a}}]$ through $[M_{\text{d}}]$ are the transfer matrix of the individual Helmholtz resonator cell and $[N_0]$ is the transfer matrix relating the Helmholtz resonator cells and the waveguide. The individual transfer matrix of the Helmholtz resonator can be tuned by adjusting the geometry. The transfer matrices of the Helmholtz resonator cells (e.g., cell $a$) and $N_0$ can be written as:

$$[M_{\text{a}}] = \begin{bmatrix} \frac{2-\alpha_{\text{a}}}{2} & \frac{-\alpha_{\text{a}}}{2} \\ \frac{\alpha_{\text{a}}}{2} & \frac{2+\alpha_{\text{a}}}{2} \end{bmatrix}, \quad (10)$$

and

$$[N_0] = \begin{bmatrix} e^{jkh_1} & 0 \\ 0 & e^{-jkh_1} \end{bmatrix}. \quad (11)$$

Here, $\alpha_{\text{a}} = R_{w1}/Z_{\text{a}}$ and $R_{w1} = \rho_0 c_0/w_1$ is the acoustic impedance of the straight channel, $Z_{\text{a}}$ is the acoustic impedance of the Helmholtz resonator $a$. The same approach can be applied to the resonators $b$, $c$, and $d$.

The detailed derivation of $Z_{\text{a}}$ is given in[19], and is directly given here for brevity:

$$Z_{\text{a}} = Z_{\text{n}} \frac{Z_{\text{c}} + jZ_{\text{n}} \tan(kw_2)}{Z_{\text{n}} + jZ_{\text{c}} \tan(kw_2)} + j\text{Im}(Z_d). \quad (12)$$

Here, $Z_{\text{n}} = \rho_0 c_0/h_2$ and $Z_{\text{c}}$ are the acoustic impedance of the neck and the cavity of the Helmholtz resonator, respectively. $\text{Im}(Z_{\text{d}})$ is the radiation impedance, which is expressed as:

$$Z_{\text{d}} = \frac{\rho_0 c_0}{w_1 h_2^2} \frac{1 - e^{-jkh_2} - jkh_2}{k^2} + \frac{\rho_0 c_0}{w_1 h_2^2} \sum_{n=1} \frac{1 - e^{-jk'_{zn}h_2} - jk'_{zn}h_2}{k_{zn}^{'3}} \quad (13)$$

with $k'_{zn} = \sqrt{k^2 - k_{xn}^2}$ and $k'_{xn} = n\pi/w_1$. The acoustic impedance of the cavity $Z_{\text{c}}$ is

given by:

$$Z_{\text{c}} = \sum_n \rho_0 c_0 \frac{k\left(1 + e^{2jk''_{xn}w_3}\right)\Phi_n^2}{k''_{xn}h_3\left(1 - e^{2jk''_{xn}w_3}\right)}. \quad (14)$$

where $\Phi_n = \sqrt{2 - \delta_n} \cos(n\pi/2)\text{sinc}(n\pi h_2/2h_3)$ and $k''_{xn} = \sqrt{k^2 - (n\pi/h_3)^2}$.

The transfer matrices of $[M_{\text{in}}]$ and $[M_{\text{out}}]$ are expressed as:

$$[M_{\text{in}}] = \begin{bmatrix} \frac{1}{2} & \frac{R_{w1}}{2} \\ \frac{1}{2} & -\frac{R_{w1}}{2} \end{bmatrix}, \quad (15)$$

and

$$[M_{\text{out}}] = \begin{bmatrix} 1 & 1 \\ \frac{1}{R_{w1}} & -\frac{1}{R_{w1}} \end{bmatrix}. \quad (16)$$

By inserting Eqs. (12), (13), and (14) into Eq. (10), the transfer matrix of an individual Helmholtz resonator unit can be obtained, which can further be combined with Eq. (9) to compute the total transfer matrix. The total transfer matrix can be tuned by adjusting the geometrical values of the unit cell. In our design, $w$, $w_2$, $h_1$, $h_2$, $h_3$ are fixed, the heights of the Helmholtz resonator cells $w_{a,b,c,d}$, and channel $w_1$ are put in the genetic algorithm to for the computation of the optimized structure.

Once the transfer matrix has been calculated, we can directly calculate the corresponding impedance matrix as

$$\begin{bmatrix} Z_{11} & Z_{12} \\ Z_{21} & Z_{22} \end{bmatrix} = w \begin{bmatrix} \frac{M_{11}}{M_{21}} & \frac{M_{11}M_{22} - M_{21}M_{12}}{M_{21}} \\ \frac{1}{M_{21}} & \frac{M_{22}}{M_{21}} \end{bmatrix}. \quad (17)$$

These expressions have been used for calculating the actual impedance values in Fig. 3d. Furthermore, we can calculate the scattering matrix as

$$\begin{bmatrix} r^+ & t^- \\ t^+ & r^- \end{bmatrix} = \begin{bmatrix} \frac{(Z_{11}-Z_0)(Z_{22}+Z_0)-Z_{21}Z_{12}}{\Delta Z} & \frac{2Z_{12}Z_0}{\Delta Z} \\ \frac{2Z_{21}Z_0}{\Delta Z} & \frac{(Z_{11}+Z_0)(Z_{22}-Z_0)-Z_{21}Z_{12}}{\Delta Z} \end{bmatrix}, \quad (18)$$

where $\Delta Z = (Z_{11} + Z_0)(Z_{22} + Z_0) - Z_{21}Z_{12}$. This equation has been used for calculating the scattering coefficients represented in Fig. 2. Comparison between this method and numerical simulations is presented in Supplementary Note 5.

**Genetic algorithm.** In the optimization, we used a genetic algorithm with continuous variables to find the optimized parameters for the resonators. The population size is 10 and the mutation rate is 0.2. We kept half of the genes for every generation and the best one does not mutate. There is no crossover in the optimization process. The optimization stops after 1500 generations. The algorithm is run 50 times for each cell to find the best match. In the design of 70° and 80° refraction, we used COMSOL Livelink with MATLAB to calculate the structures' impedance matrices. Details about the convergence of the optimization process summarized in Supplementary Note 6.

It is noted that the structure can be further optimized locally with simulations. This can be implemented by setting the pre-optimized parameters as the first generation. All the genes will be similar to the first generation and evolve gradually through mutation and crossover to search for the optimum gene locally.

**Theoretical requirements for a scattering-free metasurface.** In the theoretical derivation of the impedance profile in Results, we consider the following incident and transmitted pressure fields

$$p^+(x, y) = p_0 e^{-jk(\sin\theta_i x + \cos\theta_i y)}, \quad (19)$$

and

$$p^-(x, y) = Tp_0 e^{-jk(\sin\theta_t x + \cos\theta_t y)}, \quad (20)$$

where $p_0$ is the amplitude of the incident plane wave, $T$ is the transmission coefficient, and $\theta_{i,t}$ are the angles of incidence and refraction. The velocity fields can be written as

$$\mathbf{v}^+(x, y) = \frac{p^+(x, y)}{Z_0}[\sin\theta_i \mathbf{x} + \cos\theta_i \mathbf{y}] \quad (21)$$

and

$$\mathbf{v}^-(x, y) = \frac{p^-(x, y)}{Z_0}[\sin\theta_t \mathbf{x} + \cos\theta_t \mathbf{y}]. \quad (22)$$

For optimal performance of the metasurface, all the incident energy has to be redirected to the desired direction by a scattering-free metasurface. This condition,

equivalent to energy conservation in the normal direction in all the point of the metasurface, $\mathbf{n} \cdot \mathbf{I}^{+}(x, 0) = \mathbf{n} \cdot \mathbf{I}^{-}(x, 0)$. Therefore, the required amplitude ratio of the transmitted wave and incident wave is given by

$$T = \sqrt{\frac{\cos \theta_i}{\cos \theta_t}}. \qquad (23)$$

Expanding Eq. (1) with the assumed incident and transmitted fields, simplifying with the lossless and passive assumptions, $Z_{ij} = jX_{ij}$, and defining $\Phi_x = k(\sin \theta_i - \sin \theta_t)$, the following relations are obtained by equating both the real and imaginary parts in the equation:

$$1 = T \frac{\cos \theta_t}{Z_0} \sin(\Phi_x x) X_{12} \qquad (24)$$

$$0 = \frac{\cos \theta_i}{Z_0} X_{11} - T \frac{\cos \theta_t}{Z_0} \cos(\Phi_x x) X_{12} \qquad (25)$$

$$T \cos(\Phi_x x) = T \frac{\cos \theta_t}{Z_0} \sin(\Phi_x x) X_{22} \qquad (26)$$

$$T \sin(\Phi_x x) = \frac{\cos \theta_i}{Z_0} X_{12} - T \frac{\cos \theta_t}{Z_0} \cos(\Phi_x x) X_{22} \qquad (27)$$

Putting the energy constraint shown in Eq. (23) into Eq. (27), all the components of the impedance matrix can be obtained, yielding Eq. (5).

**Numerical simulations**. The full wave simulations based on finite element analysis are performed using COMSOL Multiphysics Pressure Acoustics module, where a spatially modulated Gaussian wave is incident normally on the metasurface. Perfectly matched layers are adopted to reduce the reflection on the boundaries. The loss in the air is modeled by the viscous fluid model in the Pressure Acoustic Module, with dynamic viscosity of $1.82 \times 10^{-5}$ Pa · s and bulk viscosity of $5.46 \times 10^{-2}$ Pa · s. The efficiencies are calculated by extracting the amplitude of the wave in the designed direction and then calculate the energy flow. The normal component is then calculated to compute the efficiency.

**Experimental apparatus**. The samples were fabricated with fused deposition modeling three-dimensional printing. The printed material is acrylonitrile butadiene styrene plastic with density of 1180 kg/m³ and speed of sound 2700 m/s, making the characteristic impedance much larger than that of air, and the walls are therefore considered to be acoustically rigid. The fabricated metasurface consists of nine periods and is secured in a two-dimensional waveguide for the measurement. A loudspeaker array with 28 speakers sends a Gaussian modulated beam normally to the metasurface and the transmitted field is scanned using a moving microphone with a step of 2 cm[48]. The acoustic field at each spot is then calculated using Fourier Transform. The overall scanned area is 114 cm by 60 cm and the signal at each position is averaged out of four measurements to reduce noise. The efficient is calculated by performing Fourier Transform along a line right behind the metasurface. The energy portion of each wave vector $k_x$, is then normalized, as shown in Fig. 5b.

**Data availability**. The data that support the findings of this study are available from the corresponding authors upon reasonable request.

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

## Acknowledgements

This work was supported by the Multidisciplinary University Research Initiative grant from the Office of Naval Research (N00014-13-1-0631), an Emerging Frontiers in Research and Innovation grant from the National Science Foundation (Grant Number 1641084), and in part by the Academy of Finland (project 287894 and 309421).

## Author contributions

S.A.C. and S.A.T. supervised the project. J.L., C.S., and A.D.-R. developed the mathematical model. J.L. performed simulations and fabricated the samples. J.L. and C.S. realized the experiments. All authors contributed to discussing the results and preparing the manuscript.

## Additional information

**Competing interests:** The authors declare no competing interests.

