## [Peer Review File(PDF 1174 kb) · Nature Communications]

Reviewers' comments:

Reviewer #1 (Remarks to the Author):

The authors present a numerical and experimental study of a reflection-less acoustic metasurface, capable of redirect the incoming energy along the desired direction (the deflection angle). The reflection-less condition is achieved by means of bianisotropy, which is implemented using a non-symmetric distribution of resonators along the thickness of the metasurface. Three deflection angles are studied numerically: 60,70 and 80 degrees, and a good agreement is found with the theoretically designed functionality. Finally, experiments are performed for the 60 degrees case, with an excellent agreement with the theory.

The paper is interesting and the results consistent with the theory, although there are some points that have to be clarified before the paper be accepted for NC.

Comment 1: Figure 2 Looks like confusing for the reader. The two panels are labelled as Figure 2(a) and Figure 2(d). I guess there is a mistake and should be Figure 2(b). In Figure 2a the incident field has the subindex "i" in the upper panel and "I" (capital) in the lower one. It seems from the figure that the field enters to the resonator through its wall, while propagation of sound happens only above the resonators. Finally, the signs +/- in the definitions of the fields are not consistent with equation (2), perhaps the transmitted fields in both panels require of a different sign? All the fields in the upper panel are labelled +, and in the lower panel are labeled -. This seems to be inconsistent with equation 2, since for any P_i equal to zero both signs are excited.

Comment 2: In Figure 3, subplots are labeled in an unnatural order as they are mentioned in the caption: b,d and f and later a,c, and d. Why not label the upper panels a,b,c and the lower panels, d,e and f? The caption should appear more natural after that.

Comment 3: The authors define the efficiency of the metasurface as stating that all the incoming energy is redirected to the desired direction. They state that for the deflection angles of 70 and 80 the efficiencies are of 96% and 91%, however in figure 3, it seems that a strong reflected field is excited, could the authors clarify this? It should be interesting to see the amplitude of both the reflected and transmitted fields, as they did in figure 5b.

Comment 4: Last sentence of page 3. I'd say that the condition $t_+ = t_- = t$ is general, not only for lossless and reciprocal particles. Also, the sentence is broken as "...and reflection coefficient..." and continues in the other side of the page as " $|t|^2 + |r|^2 = 1$ ". It seems that the full sentence is not well written. I guess that the authors want to explain that in general $t_+ = t_- = t$, but that the reflection coefficients are equal in phase and modulus for lossless and reciprocal particles, only equal in modulus for lossless but bianisotropic materials, and different in both phase and modulus for lossy and bianisotropic materials. The fact that the phases of the reflection coefficient are different but not their modulus confirm their hypothesis of having a bianisotropic lossless interface. Also, the true condition for energy conservation in the lossless case should be $|t|^2 + r^+ r^{*-} = 1$.

Comment 5 :In figure 5, after explaining panels a and b, the authors explain panel c, but they give two references to panel b...and do not mention panel d. I guess it is a general mistake in the caption.

Comment 6: In figure 5, panel b, are the labels of the diffraction orders correct? If the overall structure is a diffraction grating of period D, and the incidence of the field is normal to the metasurface, I'd expect that the diffracted modes appear at $\pm 2\pi n/kD$, so that the metasurface is designed in such a way that only the $n=-1$ order be relevant, being negligible the amplitud of the

others. That is why there is a peak at the $k_x/k = -1$ in the figure, but I'd say that the zero order is the "ballistic" peak at $k_x/k = 0$. Indeed, we can see some peaks around the positions $k_x/k = +/- 1, +/- 2, +/- 3$, for clarity, these peaks have to be labelled as the corresponding diffraction orders, and state that it is the $n = -1$ the desired one.

Comment 7: The metasurface has a period $D = 13.2\text{cm}$, which is designed so that for the frequency of 3kHz the first diffracted mode appears at an angle of $\pi/3$, so that they have used a wavelength for the field of 11.43. It means therefore that the second diffraction mode should appear at an angle given by $\sin\theta_n = 2 * 11.43 / 13.2 = 1.7318$, which is higher than one and therefore evanescent, as will be all the other modes. Therefore I assume that all the peaks found in figure 5b for k_x/k higher than one are actually evanescent modes. Why they appear in that figure? Is it because the figure is done analyzing the data too close to the metasurface? or it is because the beam used in the experiments is not exactly a plane wave and has other "incident angles"? Could the authors clarify the origin of these diffraction orders?

Comment 8: In section "Design of acoustic bianisotropic gradient metasurfaces", the reader would acknowledge a summary of the dimensions of the metasurface in terms of the operating wavelength, that is, λ/D and λ/w .

Comment 9: Same section as before, the design process is not well summarized, the definition of the macroscopic transmission coefficient T as some quantity higher than 1 can be confusing according to what typically is understood by transmission coefficient. Equations 1 to 3 should be better explained or at least a reference given.

Reviewer #2 (Remarks to the Author):

[Please see the following page.]

This paper describes bianisotropic acoustic metasurface structure for high efficiency anomalous refraction, especially for large transmission angle. For this purpose, authors proposed/used a bianisotropic acoustic cell composed of cascaded Helmholtz resonators. Experimental demonstration of scattering-free acoustic wave steering, to 60, 70 and 80 degrees of transmission angle for normally incident wave was made.

First of all, this reviewer found that there exist large mismatch between the title and contents.

As described in authors' abstract, introduction, and throughout the manuscript, this work focuses on acoustic cell architecture and its design.

(Abstract) In this work, we propose and experimentally verify *the use of a new acoustic cell architecture* that provides enough degrees of freedom to fully control the bianisotropic response and minimizes the implementation losses produced by resonant elements. **(Introduction)** In this work, a versatile platform for bianisotropic metasurfaces *based on the use of four independent resonators* is developed.

In contrast, the title of this paper claims the "Bianisotropic metasurfaces for scattering-free manipulation", which have been theoretically studied before both in electromagnetics and acoustics domain, for reflection and transmission waves.

In this sense, the scope of the claim is much narrower than its title. The achievement in this manuscript within the scope of above discussion, then is considered as; the "experimental" realization of bianisotropic metasurface.

Even in terms of mentioned "experimental realization of acoustic bianisotropic metasurface", this reviewer found that the situation is little different from authors' below claim.

(Introduction) *Although bianisotropy in acoustics, ..., has been reported recently in a single cell [39, 40], the integration of bianisotropy into a macroscopic acoustic metasurface for perfect wavefront modulation with controlled asymmetric response is not reported.*

In fact, in [39], this reviewer found the "experimental" demonstration of bianisotropic "metasurface" for transmission / reflection wavefront modulation, with the control of local scattering amplitude / phases. Authors' claim of "first experimental implementation of bianisotropic metasurface" is therefore not correct.

In this sense, the scope of authors' work gets smaller even further. The net achievement is considered as "experimental verification of former theory - minimization of scattering loss at high angle" with the implementation of "specific bianisotropic acoustic cell architecture".

Therefore, this reviewer conclude that the current submission does not hold sufficient novelty (in terms of importance, generality, innovation), necessary to justify the publication in Nature Communications. This reviewer thus suggest authors to submit their results to other journal of practical / archive nature.

Some technical comments and suggestions are listed below:

-In Fig. 1 (b), the efficiency of GSL designs and bianisotropic designs are lower than theoretical limits (possibly the result of discretization?). Clarify the reason of this penalty. This reviewer also recommend to describe the details of procedure, for the calculation/measurement of efficiencies.

- The authors described that, four-cascade Helmholtz structures are used to avoid resonance frequency range (of large loss), while three-cascade structure can also provide arbitrary bianisotropic responses. However, such increase of metamaterial unit cell (Helmholtz resonator) numbers naturally accompanies increased thickness of the metasurface. When the thickness of three unit-cell structure already exceeds a half wavelength, any attempt further increasing the thickness of "metasurface" need to be rigorously justified.
- As this paper emphasizes a non-resonance system for the mitigation of absorption, this reviewer want to see the systematic discussion on the effect of additional resonator to the loss. In detail, it seems that elongated propagation path length and redundant resonators are contributing to the reduction of *impedance* requirement for *each* resonator, consequently resulting in the reduced *loss* values in the non-resonant frequency region (in the view of Kramers-Kronig relations). Corresponding discussions will be appreciated.
- In order to justify the use of additional (fourth) resonator, this reviewer also suggest to include the required / achievable *complex* impedance ranges of single Helmholtz resonator (Z_1, Z_2, Z_3, Z_4), and how they are related to the impedance matrix requirements (Z_{11}, Z_{12}, Z_{22}), like equations S3-S5 and Fig. S2 (b) in the Supplementary Note 1. This is important not only to analyze composite structure, but also to verify the result of optimization made by genetic algorithm.
- Possibility of local-optimum design with the use of genetic algorithm should be discussed.
- In Fig. 2 (a), parameters h_1, h_2 and w_2 are missing.
- There is no Fig.2 (b)-(c) at all (not called in the text, caption either). Fig.2(d) must be Fig.2 (b).
- In caption of Fig. 3. second line, (d) seems like the errata of (e).
- In Fig. 3, this reviewer suggest to include zoom-in image of field patterns near the metasurface (it is hard to see the phase evolution at current stage). Possibly, describe (add plots for) designed values of forward/backward reflection and transmission coefficient for each cells, for better comprehension of readers.
- 70 and 80 degrees scattering simulations show power transmission efficiencies of 96% and 91% respectively. However, the ripple patters in Fig.3 (c) and (e) (for bianisotropic metasurface) suggest the existence of non-negligible back-scatterings, especially when compared to the case of Fig.3 (a) (efficiency 93%). Please provide proper explanation.
- In Fig. 4, unit of the resonance frequency is missing (5kHz?).
- In Fig. 5, experimental data is not shown in the left side of reflection domain. Explain why.
- Macroscopic transmission coefficient T_r could be confused with transmission coefficient t if used without any definition. To improve readability, define it properly. Also, consider changing the variable A in Fig.1(a) to T_r for consistency.

Reviewer #3 (Remarks to the Author):

This manuscript presents the first known experimental demonstration of the use of acoustic bianisotropy to significantly improve transmission efficiency of gradient metasurfaces when compared with metasurfaces that do not employ bianisotropic behavior. The work is scientifically sound and well presented. Due to the fact that it is the first to make use of bianisotropy outside of the seminal validating works by Koo et al (ref [39] of the manuscript) and Muhlestein et al (ref [40] of the manuscript), it is likely to be highly cited by others seeking to improve the performance of acoustic metamaterial devices. Based on this, I recommend it for publication once the following minimal points are addressed.

1. The authors correctly note that the unit cell primitive shown in Fig. 2a will generate bianisotropic effects in the direction indicated by the direction of propagation of the incident, reflected, and transmitted waves. However, there is also asymmetry in the orthogonal direction, and that bianisotropy is not measured or accounted for. This should be mentioned and the limitation to the case shown in this work should be justified.
2. The method used to extract the bianisotropic effects differs from that of ref [40], which maintained the boundary condition at the termination of the impedance tube. The reason/justification of using the approach described in this work as opposed to that described in [40] should be provided.
3. Note that Koo et al (ref [39]) used multiple unit cells with membranes to demonstrate bianisotropy, as opposed to a single unit cell as described in the current manuscript. The present manuscript states that [39] only shows the case of a single unit cell and seems to suggest that only [28] proposed use of membranes to achieve acoustic bianisotropy, which is not the case. Please correct this in the resubmission.
4. While the authors do seem to be aware of the relevant previous work on this topic, more clear emphasis should be placed on the enabling experimental demonstrations of refs [39] and [40] and detailed analogies between the electromagnetic and acoustic cases described by [38]. The idea of acoustic bianisotropy was not of clear utility or interest until only very recently, and it was those works that showed formal analogies as well as experimental demonstrations of the behavior.

Summary of changes

1. One of the biggest concerns from the reviewers is that we need to clarify our contribution to this field. To address this concern, we have changed the title and introduction to more clearly reflect the two key elements of our contribution: first, our multi-resonator design approach for achieving the needed bianisotropy of the individual cells, and thus a systematic and practical design approach for metasurfaces as long as the input and desired output fields on both sides are known; and second, our experimental demonstration of wavefront shaping using these bianisotropic unit cells that is significantly more efficient than what can be achieved with simple phase control.

2. We have also added comments about the relationship of our work to the existing literature, showing the clear difference between our work and Ref. [39], as the reviewer suggested. To make it clear, we added a new section in the supplementary material to discuss the local scattering properties and the controlling abilities for the meta-atoms and the situations where bianisotropy is need. In the main text we also largely modified the introduction part to define and distinguish the contributions of the previous work and our work to the field of wave control.

3. Referee #2 is interested in discussion of the comparison between implementing the required unit cell with 3 resonators and our implementation with 4 resonators. To address this concern, we tried to design a unit cell with 3 resonators that has the same impedance response as a cell with 4 resonators. We have found that: first, designing the cell within the same dimension constraints using 3 resonators is not possible because the required resonances cannot fit into the available physical space; second, after releasing some constraints in geometry, the unit cell can be designed with 3 resonators, but the structure was found to be more sensitive to loss because the individual resonators must operate closer to resonance. Moreover, the 3 resonator design is much more sensitive to fabrication error since change of acoustic properties are drastic near resonant frequencies, and it may cause difficulty in fabrication because of smaller sizes of the necks. This detailed discussion and comparison is added to the supplementary material.

We also added a deeper theoretical discussion of the four resonator approach and show the necessity of applying optimization for the practical designs.

Comments and main conclusions regarding these discussions are also added to the main text.

4. One of the concerns is that the plotted fields contain reflected waves. We have added a paragraph with careful calculation clarify that they do not carry significant energy despite their nonnegligible pressure amplitude. Therefore, the overall energy efficiency is not significantly affected by these reflected modes.

5. We added a complete derivation of transmission and reflection coefficients for bianisotropic meta-atoms. All the transmission and reflection coefficients of the unit cells are calculated and provided in the tables in supplementary materials.
6. The detailed procedure for the calculation/measurement of efficiencies are added in Numerical simulation and Experimental apparatus subsections in the Method section.
7. The zoom-in images of the fields are added to the revised Figure 3 to show the phase evolution.
8. Justification of the proposed method for retrieving the acoustic properties in simulation is provided. Also, we compared our method and the method provided in ref. [40], and showed that these two methods are mutually equivalent.
9. The asymmetry in the orthogonal direction is discussed in the revised paper. Based on the discussion, limitations to the designing process is further discussed.
10. The dimensions of the metasurface is summarized in terms of operating wavelength.
11. The definitions of all the signs, variables and subscripts are checked and explained carefully.
12. The labels, captions and ordering of the subfigures are examined and adjusted to make them referred to in a more logical way.
13. Correction of typos.

Response to Reviewer #1

The authors present a numerical and experimental study of a reflection-less acoustic metasurface, capable of redirect the incoming energy along the desired direction (the deflection angle). The reflection-less condition is achieved by means of bianisotropy, which is implemented using a non-symmetric distribution of resonators along the thickness of the metasurface. Three deflection angles are studied numerically: 60, 70 and 80 degrees, and a good agreement is found with the theoretically designed functionality. Finally, experiments are performed for the 60 degrees case, with an excellent agreement with the theory.

The paper is interesting and the results consistent with the theory, although there are some points that have to be clarified before the paper be accepted for NC.

Response: We wish to thank the referee for the careful review and positive remarks on our work.

Comment 1: *Figure 2 Looks like confusing for the reader. The two panels are labelled as Figure 2(a) and Figure 2(d). I guess there is a mistake and should be Figure 2(b). In Figure 2a the incident field has the subindex "i" in the upper panel and "I" (capital) in the lower one. It seems from the figure that the field enters to the resonator through its wall, while propagation of sound happens only above the resonators. Finally, the signs +/- in the definitions of the fields are not consistent with equation (2), perhaps the transmitted fields in both panels require of a different sign? All the fields in the upper panel are labelled +, and in the lower panel are labeled -. This seems to be inconsistent with equation 2, since for any P_i equal to zero both signs are excited.*

Response: We apologize for the mistake in labeling. In the revised version, we have changed the label for Figure 2(d) to Figure 2(b).

In Figure 2(a) the capital "I" has been changed to the lower case.

We moved the arrows in Figure 2(a) further from the resonators, so that it doesn't look like the sound goes into the structure from the walls. The colors of the arrows are also modified so that they correspond to the representations in Fig. 2(b).

The definition of the +/- signs in Figure 2 and equations denote different field components. As stated in the paper, the signs in Figure 2 refers to whether the incident wave is from positive or negative direction, while in all the equations, the signs refer to the fields at both sides of the metasurface. The reason why we set the definitions is that we need all the definitions of the transmission and reflection coefficients to be consistent. To avoid confusion, we revised sentence below equation 2 to define the p_s^+ and p_s^- . The revised sentence reads: " p_{\pm}^{\pm} is the amplitude of the scattered fields at both sides of the particle (i.e. $p_{\pm}^{\pm} = p_{\pm}^{\pm} + p_{\pm}^{\pm}$ and $p_{\pm}^{\pm} = p_{\pm}^{\pm} - p_{\pm}^{\pm}$)".

Comment 2: *In Figure 3, subplots are labeled in an unnatural order as they are*

mentioned in the caption: b,d and f and later a,c, and d. Why not label the upper panels a,b,c and the lower panels, d,e and f? The caption should appear more natural after that.

Response: We set the label this way since we wanted to show the link between upper and lower panels. In the revised version, we reordered the labels in Figure 3 to make them look more natural. We also changed the caption accordingly so that the figures are referred to in a more logical way.

Comment 3: *The authors define the efficiency of the metasurface as stating that all the incoming energy is redirected to the desired direction. They state that for the deflection angles of 70 and 80 the efficiencies are of 96% and 91%, however in figure 3, it seems that a strong reflected field is excited, could the authors clarify this? It should be interesting to see the amplitude of both the reflected and transmitted fields, as they did in figure 5b.*

Response: The referee raised a very good point. When we calculate the energy efficiency of a metasurface, the conserved quantity is the power flow normal to the metasurface. On the incident side, the wave is coming from the normal direction to the metasurface, but the scattered fields have large oblique angles, which means they only carry a very small portion of the incident power flow even though they may have large field amplitude. In other words, although their amplitude appears to be large, they do not contribute much to the energy budget of the overall metasurface. Taking the 70 degree case as an example, the amplitude of the incident wave is 1 Pa and the reflected wave amplitude is 0.35 Pa. But since the reflected wave propagates at an angle of 70 degrees, the normally directed power flow out of the metasurface is only $(0.35)^2 \cos(70^\circ) = 4\%$, which does not significantly affect the energy efficiency. This is also the reason why when we bend the incident wave to a large angle, the transmitted amplitude is necessarily larger than the incident amplitude.

We added a comment in the revised paper to clarify this. The field amplitudes of the reflected waves are also given in the revised manuscript. The revised sentence reads: “The imperfect scattered field is caused by the reflection from the metasurface, which is due to non-ideal implementation of the metasurface. We note that, however, the power flow normal to the surface (the conserved quantity that defines energy efficiency) in these reflections is low and contributes little to the overall energy efficiency of the metasurface. Because the deflection angles, i.e., 70° and 80°, are large, the reflection amplitudes of 0.35 and 0.6, respectively, their contributions to normally directed power follow is only 4% and 6%. In other words, the high efficiency is still maintained even though the reflected field amplitudes are not negligible.”

Comment 4: *Last sentence of page 3. I'd say that the condition $t^+ = t^- = t$ is general, not only for lossless and reciprocal particles. Also, the sentence is broken as "...and reflection coefficient..." and continues in the other side of the page as " $|t|^2 + |r|^2 = 1$ ". It seems that the full sentence is not well written. I guess that the authors want to explain that in general $t^+ = t^- = t$, but that the reflection coefficients are equal in phase and modulus for lossless and reciprocal particles, only equal in modulus for lossless*

but bianisotropic materials, and different in both phase and modulus for lossy and bianisotropic materials. The fact that the phases of the reflection coefficient are different but not their modulus confirm their hypothesis of having a bianisotropic lossless interface. Also, the true condition for energy conservation in the lossless case should be $|t^+|^2 + |r^+|^2 + |r^-|^2 = 1$.

Response: We thank the reviewer for this comment. We agree with the referee that the condition $t^+ = t^-$ is general for not only for lossless particles. In that sentence we meant to say that they satisfy both conditions in lossless and reciprocal cases. We apologize for our typo. To avoid the ambiguity, we changed the sentence as: “For lossless and reciprocal particles, the transmission coefficients and reflection coefficients satisfy $t^+ = t^-$, $|t^+|^2 + |r^+|^2 + |r^-|^2 = 1$, and $r^+ + t^+ + r^- = 0$ ”.

We consider that this general discussion would be valuable for the readers and we have included a more detailed explanation of the conditions for lossless and reciprocal metasurfaces in the Supplementary Material (see Note 1). These conditions are summarized in Eqs. S3 and S4. They are different to the one suggested by the reviewer ($|t^+|^2 + |r^+|^2 + |r^-|^2 = 1$). We suspect that the condition suggested by the reviewer is not generally true because one can find cases in which it is not satisfied. For example, if we consider a bianisotropic material that is made of a slab of hard wall ($r=1$) and a slab of vacuum ($r=-1$) this statement is not correct.

Comment 5: *In figure 5, after explaining panels a and b, the authors explain panel c, but they give two references to panel b...and do not mention panel d. I guess it is a general mistake in the caption.*

Response: We apologize for the typo. The caption is revised as:” (a) Schematic representation of the experimental setup and a period of the fabricated sample. (b) Comparison between the normalized scattering of the bianisotropic metasurface (experimental and numerical) and a GSL design. (c-d) Analysis of the real part (c) and magnitude square (d) of the experimental pressure field and the comparison with the numerical simulations.”

Comment 6: *In figure 5, panel b, are the labels of the diffraction orders correct? If the overall structure is a diffraction grating of period D , and the incidence of the field is normal to the metasurface, I'd expect that the diffracted modes appear at $\pm 2\pi n/kD$, so that the metasurface is designed in such a way that only the $n=-1$ order be relevant, being negligible the amplitud of the others. That is why there is a peak at the $k_x/k=-1$ in the figure, but I'd say that the zero order is the "ballistic" peak at $k_x/k=0$. Indeed, we can see some peaks around the positions $k_x/k=\pm 1, \pm 2, \pm 3$, for clarity, these peaks have to be labelled as the corresponding diffraction orders, and state that it is the $n=-1$ the desired one.*

Response: The referee raised a very good point and offered us a new perspective to understand the metasurface. Indeed, if we look at it from the grating perspective, the orders that matters should be the -1 th order. As pointed out by the referee, we can see some peaks around the positions $k_x/k=\pm 1, \pm 2, \pm 3$. In the revised figure, we labelled the peaks as the corresponding

diffraction orders.

The sentences referring to Figure 5(b) are also revised accordingly. For example, “The experimental results show that unwanted diffraction orders are greatly suppressed”; “most of the energy localized in the desired direction (-1 order) and other diffraction modes are severely suppressed”; “...is also shown for comparison where undesired diffraction orders can be clearly observed”

Comment 7: *The metasurface has a period $D=13.2\text{cm}$, which is designed so that for the frequency of 3kHz the first diffracted mode appears at an angle of $\pi/3$, so that they have used a wavelength for the field of 11.43 . It means therefore that the second diffraction mode should appear at an angle given by $\sin\theta_n=2*11.43/13.2=1.7318$, which is higher than one and therefore evanescent, as will be all the other modes. Therefore I assume that all the peaks found in figure 5b for k_x/k higher than one are actually evanescent modes. Why they appear in that figure? Is it because the figure is done analyzing the data too close to the metasurface? or it is because the beam used in the experiments is not exactly a plane wave and has other "incident angles"? Could the authors clarify the origin of these diffraction orders?*

Response: The referee is correct that these nonpropagating modes appear because the fields are analyzed very close to the surface, and this issue needs to be clarified. As stated in the paper, “the normalized energy distribution on each direction is further calculated by performing the Fourier transform along the line right behind the metasurface”. They appear in that figure because the data is captured right behind the metasurface and the evanescent modes are observed.

To make it clear, we revised the sentence as “most of the energy localized in the desired direction (-1 order) and other diffraction modes are severely suppressed, including the high-order evanescent modes.”

Comment 8: *In section "Design of acoustic bianisotropic gradient metasurfaces", the reader would acknowledge a summary of the dimensions of the metasurface in terms of the operating wavelength, that is, λ/D and λ/w .*

Response: We would like to thank for the referee’s suggestion. In the revised version, we added the relative dimension of the structure to the sentence, which reads: “The operating frequency is chosen to be 3000 Hz that makes the period of the metasurface $D=13.2\text{ cm}$ ($D/\lambda=1.15$). We use 11 cells along the period for implementing the spatial dependent bianisotropic response, so the width of the unit cell is $w=D/11=12\text{ mm}$ ($w/\lambda=0.10$).”

Comment 9: *Same section as before, the design process is not well summarized, the definition of the macroscopic transmission coefficient T as some quantity higher than 1 can be confusing according to what typically is understood by transmission coefficient. Equations 1 to 3 should be better explained or at least a reference given.*

Response: In the previous draft we defined the macroscopic transmission coefficient T with Eqn. (21) in the Method section, and detailed information about Eqn. (1-3) is also given in the same

section by Eqn. (17-25). T is defined as the pressure amplitude transmission coefficient. It is important to note that this standard definition of T does not directly connect to power or conservation of power, which also fold in the angle of incidence. In the revised paper, we clarify this point with " This condition, equivalent to energy conservation in the normal direction, requires the macroscopic pressure transmission coefficient to satisfy $T=1/\sqrt{\cos\theta_{\text{t}}}$. Detailed definition of the macroscopic transmission coefficient can be found in the Method section."

Response to Reviewer #2

This paper describes bianisotropic acoustic metasurface structure for high efficiency anomalous refraction, especially for large transmission angle. For this purpose, authors proposed/used a bianisotropic acoustic cell composed of cascaded Helmholtz resonators. Experimental demonstration of scattering-free acoustic wave steering, to 60, 70 and 80 degrees of transmission angle for normally incident wave was made. First of all, this reviewer found that there exist large mismatch between the title and contents.

As described in authors' abstract, introduction, and throughout the manuscript, this work focuses on acoustic cell architecture and its design.

(Abstract) In this work, we propose and experimentally verify the use of a new acoustic cell architecture that provides enough degrees of freedom to fully control the bianisotropic response and minimizes the implementation losses produced by resonant elements.

(Introduction) In this work, a versatile platform for bianisotropic metasurfaces based on the use of four independent resonators is developed.

In contrast, the title of this paper claims the "Bianisotropic metasurfaces for scattering-free manipulation", which have been theoretically studied before both in electromagnetics and acoustics domain, for reflection and transmission waves.

In this sense, the scope of the claim is much narrower than its title. The achievement in this manuscript within the scope of above discussion, then is considered as; the "experimental" realization of bianisotropic metasurface.

Response: We wish to thank the referee for offering careful and constructive advice on our work that has helped us improve our manuscript. In order to make the title to be more specific and informative, we have changed the title to be "Systematic design and experimental demonstration of bianisotropic metasurfaces for scattering-free manipulation of acoustic wavefronts". This title better reflects the two key novel elements of our contribution: first, our the concept and validation of our multi-resonator topology for achieving the needed acoustic bianisotropy response with a reasonable topology for the implementation of gradient metasurfaces; and second, the first experimental demonstration of perfect wavefront shaping using these bianisotropic unit cells that is significantly more efficient than what can be achieved with any of already known gradient metasurfaces.

We would also like to emphasize that the impact of this manuscript goes beyond the analysis of acoustic bianisotropy, its experimental demonstration or another gradient bianisotropic metasurface. In this paper, we analyze the topology of the meta-atoms and the requirements for the design of efficient bianisotropic gradient metasurfaces. As we have explained in the new version of the manuscript, a good topology needs: small sizes (tangential direction), full control of the bianisotropy, and low losses. For the first time, in this manuscript, we deal with these three important aspects simultaneously and show a robust design strategy which can be easily applied to other application such as acoustic hologram or lenses. The versatility is clearly shown with the design and experimental demonstration of three perfect acoustic metasurfaces for anomalous refraction. For this reason, we believe that this manuscript makes an important contribution to the development of bianisotropic metasurfaces and we have revised the manuscript accordingly to

stress these aspects (green parts in the revised manuscript).

Even in terms of mentioned “experimental realization of acoustic bianisotropic metasurface”, this reviewer found that the situation is little different from authors’ below claim.

(Introduction) Although bianisotropy in acoustics, ..., has been reported recently in a single cell [39, 40], the integration of bianisotropy into a macroscopic acoustic metasurface for perfect wavefront modulation with controlled asymmetric response is not reported.

In fact, in [39], this reviewer found the “experimental” demonstration of bianisotropic “metasurface” for transmission / reflection wavefront modulation, with the control of local scattering amplitude / phases. Authors’ claim of “first experimental implementation of bianisotropic metasurface” is therefore not correct.

In this sense, the scope of authors’ work gets smaller even further. The net achievement is considered as “experimental verification of former theory - minimization of scattering loss at high angle” with the implementation of “specific bianisotropic acoustic cell architecture”.

Response: We would like to thank the reviewer for pointing this out. Here we would like to note that we “experimentally characterize the first bianisotropic gradient metasurface for perfect acoustic anomalous refraction”, while in [39], the such perfect wavefront manipulation is not achieved in either simulation or experiment. This distinction was not made clear in the previous version of the manuscript. Accordingly, in the revised manuscript, we rephrased the introduction and added more comments about [39].

We have included a new analysis of the local scattering properties of the perfect metasurface which clearly show the differences between both designs. The gradient metasurface proposed in our work is not a minor refinement of the one in ref. [39], as the physics behind both designs is completely different. In ref [39] the bianisotropy is used to split the energy (50% transmitted and 50% reflected) and control the direction with a simple phase gradient that will not provide perfect performance as was shown in [28]. We use bianisotropy for ensuring the power conservation and provide 100% of transmission in the desired direction.

Finally, we would like to stress that the design topology of the unit cells described in ref. [39] would not provide the same performance even if it were designed towards our conditions. The large size of the elements, the difficulty in precisely controlling the response of the membranes, and the inherent instability of the uniform tension in the structure will reduce the efficiency and durability in many practical scenarios. The size constraint will also apply to any other design where the gradient is sharper, and the discretization plays an important role.

Therefore, this reviewer conclude that the current submission does not hold sufficient novelty (in terms of importance, generality, innovation), necessary to justify the publication in Nature Communications. This reviewer thus suggest authors to submit their results to other journal of practical / archive nature.

Response: To the best of the authors’ knowledge, this manuscript makes two primary

contributions. The first is that we propose a **systematic design procedure of bi-anisotropic unit cells with controllable, arbitrary bi-anisotropic responses and the proper topology for implementing gradient metasurfaces**. It provides a versatile platform for almost all bi-anisotropic metasurface designs. In recent years, there has been a growing research interest in the concept of bi-anisotropic metasurface in electromagnetics, and similar concepts are beginning to emerge in the field of acoustics (so far there exists only several theoretical proposals). Since the actual implementation of such elements is the key obstacle for realizing practical applications, we believe that the systematic design of the architecture will be valuable to the acoustic community. The second contribution is that this paper proposed **the first practical design of a perfect gradient acoustic metasurface** and verified the design with both simulation and experiment. Since no explicit designs are given in former theories, we believe that our paper is not “just an experimental verification of former theory” but in fact the first realization of scattering-free manipulation using bi-anisotropic metasurfaces in the context of acoustics.

Some technical comments and suggestions are listed below:

-In Fig. 1 (b), the efficiency of GSL designs and bianisotropic designs are lower than theoretical limits (possibly the result of discretization?). Clarify the reason of this penalty. This reviewer also recommend to describe the details of procedure, for the calculation/measurement of efficiencies.

Response: The referee raised a very good point. The efficiency of GSL designs and bianisotropic designs are lower than theoretical limits mostly because of discretization of the theoretically continuous profile. If the discretization is good the efficiency will be equal to the theoretical prediction (see the figure below with the analysis of the efficiency done in [28] with a large number of elements). In the revised manuscript, we added a comment addressing this issue, which reads: " (b) Comparison of the efficiency for anomalous transmission metasurfaces. Bianisotropic designs show great advancement especially for large deflection angles. Realized efficiencies are slightly lower than the theoretical limit as a result of discretization."

In simulation, the efficiencies are calculated by extracting the amplitude of the wave in the designed direction and then calculate the energy flow normal to the metasurface. The energy flow in the desired direction is then normalized with the incident wave to compute the efficiency. In experiments, the efficient is calculated by performing Fourier Transform along a line right behind

the metasurface. The energy portion of each wave vector, k_x , is then normalized, as shown in Fig. 5(b). In the revised manuscript, we added more details, which reads:

“The efficiencies are calculated by extracting the amplitude of the wave in the designed direction and then calculate the energy flow normal to the metasurface. The energy flow in the desired direction is then normalized with the incident wave to compute the efficiency. ”

“The efficient is calculated by performing Fourier Transform along a line right behind the metasurface. The energy portion of each wave vector, k_x , is then normalized, as shown in Fig. 5(b).”

- The authors described that, four-cascade Helmholtz structures are used to avoid resonance frequency range (of large loss), while three-cascade structure can also provide arbitrary bianisotropic responses. However, such increase of metamaterial unit cell (Helmholtz resonator) numbers naturally accompanies increased thickness of the metasurface. When the thickness of three unit-cell structure already exceeds a half wavelength, any attempt further increasing the thickness of “metasurface” need to be rigorously justified.

Response: We agree with the reviewer that increasing the thickness of “metasurface” needs to be rigorously justified. However, we would also like to emphasize that the thickness of three unit-cell structure did NOT exceed a half of wavelength. Moreover, even the four-cascade Helmholtz structures still does not exceed a half of wavelength. (5 cm in total while the wavelength at operating frequency 3000Hz is 11.43 cm). We have modified the text to make clearer that the thicknesses are all less than half of a wavelength, which reads: “All the thicknesses are less than half of a wavelength.”

- As this paper emphasizes a non-resonance system for the mitigation of absorption, this reviewer want to see the systematic discussion on the effect of additional resonator to the loss. In detail, it seems that elongated propagation path length and redundant resonators are contributing to the reduction of impedance requirement for each resonator, consequently resulting in the reduced loss values in the nonresonant frequency region (in the view of Kramers-Kronig relations). Corresponding discussions will be appreciated.

Response: We thank the referee for the constructive suggestion. Indeed, from the discussion in Supplementary Note 2, the minimum requirement to achieve bianisotropy is to use three resonators. However, as we will show here, the use of only three resonators will have certain limitations in practical implementations. For example, it will induce larger error, greater instability/sensitivity to the geometry and thus fabrication error, and is therefore not employed in our design. To illustrate the necessity of using four resonators, we choose unit #1 of the 60° deflection case as an example. To replace the four-resonator cell with the three-resonator cell, the width of the cell ($w=12$ mm), thickness of the shell ($h_1=1$ mm), width of the neck ($h_2=1.5$ mm) and length of the cell (50 mm) is kept the same while the height of the channel (w_1) and the cavities ($w_{a,b,c}$) are set as variable. We run a set of Genetic Algorithm (GA) by following the same procedure with the four-resonator design to find the optimal geometry produced by using three

resonators. However, the results will not converge and an acceptable design (within 20% error) cannot be found. This is because the individual resonators cannot operate near the resonance under the given geometry (same as four-resonator designs), and without accessing the extreme values near the resonance, the resulting whole impedance matrix cannot provide the bianisotropic response required by the theory.

To release this condition, we reduce the width of the h_2 neck to 0.6 mm, so that the resonators can operate near resonance within the range of the cavity height (0 mm to 9mm). Moreover, the height of the necks of each resonator is also set to be variables so that the resulting geometry would provide more degrees of freedom. By relaxing these constraints and running GA, a design is found with a 4.47% cost. The parameters are $w_1=3.2$ mm, $w_a=4.7$ mm, $w_b=4.5$ mm, $w_c=3.6$ mm, with the height of the necks being 0.63 mm, 0.87 mm and 1.15 mm, respectively. Figure R1 shows the total acoustic field of the three-resonator cell compared with the four-resonator cell.

Figure R1. Total acoustic field of the unit cells under plane wave excitation from different directions.

It can be seen that the three-resonator cell can generally produce the same response compared with the four-resonator cell. However, one should bear in mind that such a unit cell based on three resonators is achieved by releasing some of the geometrical constraints such as the height of the necks. This will inevitably make the designing process more complicated, and the narrow neck width and varying neck height will also pose challenges to fabrication and may result in less stability and repeatability. Moreover, the small features would also make the whole structure very sensitive to fabrication errors.

In addition, we also calculate the resonance frequencies of each individual resonator in the three-resonator design, the resonance frequencies are 3.96 kHz, 3.85 kHz, and 3.97 kHz, respectively, which are all much closer to the designed operation frequency (3.0 kHz) compared to the four-resonator design. This confirms that the use of only three resonators will make the resonators work near their resonance, which will in turn increase the loss in real implementation. The use of four resonators, on the other hand, will effectively reduce the requirement for each individual resonator, making them work away from resonance, as pointed out by the referee.

We have added comments in the main text to clarify these points, which reads “However, to obtain extreme asymmetric response required by some gradient metasurfaces, the resonators will have to work near their resonant frequencies and this will make it difficult to control their responses, and also increase loss. The required resonances also puts constraints on the physical dimensions and cause robustness issues to the practical designs. In order to mitigate these practical limitations, we

propose a four side-loaded resonators particle, as shown in Fig. 2(a).”

- In order to justify the use of additional (fourth) resonator, this reviewer also suggest to include the required / achievable complex impedance ranges of single Helmholtz resonator (Z_1, Z_2, Z_3, Z_4), and how they are related to the impedance matrix requirements (Z_{11}, Z_{12}, Z_{22}), like equations S3-S5 and Fig. S2 (b) in the Supplementary Note 1. This is important not only to analyze composite structure, but also to verify the result of optimization made by genetic algorithm.

Response: The corresponding derivation for four resonators design has been added to the supplementary material Note 2. We would also like to note that, although we need at least 3 resonators in order to get enough degrees of freedom, we have also shown in the revised paper that implementation with 4 resonators provide the robustness against loss and fabrication error. However, since there are four variables and three equations, unique explicit solution cannot be found theoretically with four resonators design (in fact, there are five variables in our practical implementation since the width of the channel is another control parameter), and there will be infinite sets of solutions. In this sense, the achievable ranges of impedances under certain geometrical constraints are not trivial and we look forward to exploring in more depth in the future work.

- Possibility of local-optimum design with the use of genetic algorithm should be discussed.

Response: The reviewer raised a valid point. Further optimization at local optimum regions may be possible if we use simulations for retrieving the impedance of single cells. This can be implemented by setting the optimized parameters in the first generation, and run the optimization. All the genes will be similar to the first generation, and change randomly through mutation and crossover to search for the local optimum gene in that region.

However, local optimization using theoretically calculated impedances is generally not possible. As can be seen in Fig. 3, the errors are already very small (<1.5%) this small error will be dominated by the slight difference using the analytic model, so that further optimization will not greatly enhance the performance.

On the other hand, according to the authors’ design experience, as long as the relative errors are less than 5%, further optimization will not improve the results by a lot. Instead, further discretizing the metasurface will provide larger improvement. Furthermore, it should be emphasized that our paper does not purport to find globally optimal design solutions. Our goal is to find sufficiently good solutions that we can implement experimentally the perfect metasurface concept, and this is what we have done.

In the revised manuscript, we added the discussion about the possibility of further improving the performance by local optimization. The discussion reads: "It is note that the structure can be further optimized locally if we optimize the structure with simulations. This can be implemented by setting the pre-optimized parameters in the first generation. All the genes will be similar to the first generation, and evolve gradually through mutation and crossover to search for the local

optimum gene in that region."

- In Fig. 2 (a), parameters h_1 , h_2 and w_2 are missing.

Response: We would like to thank the referee for pointing this out. These parameters are added to the revised Fig. 2(a).

- There is no Fig.2 (b)-(c) at all (not called in the text, caption either). Fig.2(d) must be Fig.2 (b).

Response: We apologize for the typo. It should be Figure 2(b) and is corrected in the revised manuscript.

- In caption of Fig. 3. second line, (d) seems like the errata of (e).

Response: We apologize for the typo. This is corrected in the revised capital. We also reordered the label so that the figures are mentioned in a more natural way, as reviewer #1 suggested. The new capital reads: "Bianisotropic metasurfaces for scattering-free anomalous refraction. (a-c) represent the numerical simulation of the total pressure field for bianisotropic metasurfaces (left) and GSL metasurfaces (right) when $\theta_{\text{t}}=60^\circ$, 70° , and 80° . (d-f) represent the impedance matrices profile for $\theta_{\text{i}}=0^\circ$ and $\theta_{\text{t}}=60^\circ$, 70° , and 80° ."

- In Fig. 3, this reviewer suggest to include zoom-in image of field patterns near the metasurface (it is hard to see the phase evolution at current stage). Possibly, describe (add plots for) designed values of forward/ backward reflection and transmission coefficient for each cells, for better comprehension of readers.

Response: The reviewer offered a good suggestion. The zoom-in image of field patterns near the metasurface is included in the updated Fig. 3(a-c). The caption is also modified accordingly, which reads: "(a-c) represent the numerical simulation of the total pressure field for bianisotropic metasurfaces (left) and GSL metasurfaces (right) when $\theta_{\text{t}}=60^\circ$, 70° , and 80° . The insets show the phase evolution inside the metasurface. (d-f) represent the impedance matrices profile for $\theta_{\text{i}}=0^\circ$ and $\theta_{\text{t}}=60^\circ$, 70° , and 80° ."

We have added the derivation of the transmission/ reflection coefficients for an arbitrary bi-anisotropic meta-atom and added this knowledge to the main text. Please see Eqn. (6-7) and corresponding explanations in the revised manuscript.

The forward/ backward reflection and transmission coefficients for each cell are also added in the tables in Supplementary Note 6. We also added a comment about these information, which reads: "The physical dimensions of the final design and their corresponding transmission/ reflection coefficients are summarized in the Supplementary Material."

- 70 and 80 degrees scattering simulations show power transmission efficiencies of 96%

and 91% respectively. However, the ripple patterns in Fig.3 (c) and (e) (for bianisotropic metasurface) suggest the existence of nonnegligible back-scatterings, especially when compared to the case of Fig.3 (a) (efficiency 93%). Please provide proper explanation.

Response: The referee raised a very good point. When we calculate the energy efficiency of a metasurface, the most crucial thing is the energy flow normal to the metasurface. On the incident side, the wave is coming from the normal direction to the metasurface, but the scattered fields have large angles. That's why they only carry a very small portion of energy although their amplitude looks large, even comparable with the incident amplitude. Take the 70 degree case as an example, the amplitude of the incident wave is 1 Pa and the reflected wave is 0.35 Pa, which will make the reflected field seem strong. However, since the reflected wave propagates at an angle of 70 degrees, the energy it is carrying out of the metasurface is only about $(0.35)^2 \cos(70^\circ) = 4\%$, which does not affect much on the overall energy efficiency.

We added a comment in the revised paper to clarify this issue. The amplitudes of the reflected waves are also given in the revised manuscript. The revised sentence reads: "The imperfect scattered field is caused by the reflection from the metasurface, which is due to non-ideal implementation of the metasurface. We note that, however, the energy they are carrying is low and it contributes little to the overall energy efficiency of the metasurface. This is because the deflection angles, i.e., 70° and 80° , are large, although the reflection amplitudes are 0.35 and 0.6, respectively, the real energy associated with them in the normal direction is only 4% and 6%. In other words, the high efficiency is still maintained even though the fields do not look perfect."

- In Fig. 4, unit of the resonance frequency is missing (5kHz?).

Response: Figure. 4 shows the normalized frequency, i.e., $f_{a,b,c,d}/f_0$, so that it is unitless. $f_{a,b,c,d}$ are the resonance frequencies of the four individual resonators and f_0 is the operation frequency which is 3000 Hz. We can see that all of the normalized resonance frequencies are away from 1, indicating that none of the resonators is working near the resonance, and hence with less loss. To clarify this, we modified the caption in figure 4 to be "Normalized resonance frequency of the individual resonators of the scattering-free anomalous refractive metasurface designs for $\theta_{\text{in}} = 0^\circ$ and $\theta_{\text{t}} = 60^\circ$. All the resonators are working out of the resonant frequency".

We have also added a sentence to make this clearer: "Figure 4 depicts the normalized resonance frequency of each individual resonator with respect to the operation frequency f_0 (3000Hz)."

- In Fig. 5, experimental data is not shown in the left side of reflection domain. Explain why.

Response: We thank the referee for pointing this out. In Fig. 5(c, d), the left panels show the total field in simulation while the right panels show the scattered field in experiment. Since the incident and reflected waves are mainly confined in the right side of the entire reflection domain,

we only showed this region and compared it with simulation data. A box labeled “Reflection” was used to denote the reflection domain for clarity of the figure. Following the referee’s suggestions, we have updated Fig. 5 and plotted the experimental field in the entire reflection domain.

- *Macroscopic transmission coefficient T , could be confused with transmission coefficient t , if used without any definition. To improve readability, define it properly. Also, consider changing the variable A in Fig.1(a) to T , for consistency.*

Response: Thank you for pointing it out. In the original manuscript we defined and discussed the macroscopic transmission coefficient T in Eqn. (21) in the Method section, and detailed information about Eqn. (1-3) is also given in the same section by Eqn. (17-25). We apologize that we didn’t present our results in a clear way. In the revised paper, we added a sentence so that the macro transmission coefficient T is properly introduced and defined, the sentence reads: “This condition, equivalent to energy conservation in the normal direction, requires the macroscopic transmission coefficient to satisfy $T=1/\sqrt{\cos\theta_{\text{t}}}$ \$. A more rigorous definition and detailed definition of the macroscopic transmission coefficient can be found in the Method section.”

As the reviewer suggested, we have also changed the variable A in Fig.1(a) to T , for consistency.

Response to Reviewer #3

This manuscript presents the first known experimental demonstration of the use of acoustic bianisotropy to significantly improve transmission efficiency of gradient metasurfaces when compared with metasurfaces that do not employ bianisotropic behavior. The work is scientifically sound and well presented. Due to the fact that it is the first to make use of bianisotropy outside of the seminal validating works by Koo et al (ref [39] of the manuscript) and Muhlestein et al (ref [40] of the manuscript), it is likely to be highly cited by others seeking to improve the performance of acoustic metamaterial devices. Based on this, I recommend it for publication once the following minimal points are addressed.

Response: We wish to thank the referee for the careful review and positive remarks on our work.

1. The authors correctly note that the unit cell primitive shown in Fig. 2a will generate bianisotropic effects in the direction indicated by the direction of propagation of the incident, reflected, and transmitted waves. However, there is also asymmetry in the orthogonal direction, and that bianisotropy is not measured or accounted for. This should be mentioned and the limitation to the case shown in this work should be justified.

Response: The reviewer raises a valid point. Indeed, there is also asymmetry in the orthogonal direction. However, since there are walls between adjacent cells, the wave does not propagate along the orthogonal direction inside the metasurface. Therefore, we can still treat the cells individually at the design stage since the asymmetry along the orthogonal direction does not affect the wave propagation. There is also asymmetry in the single unit cell, but the asymmetry can be ignored as long as the width of the cell is much smaller than the wavelength so that only fundamental mode is allowed in the channel. In the revised manuscript, sentences are added to clarify the asymmetry along the metasurface and inside the unit cells, and limitations are discussed. The revised text is: "Although there is also asymmetry in the orthogonal direction of the unit cells, it can be ignored as long as the width of the channel is significantly smaller than a wavelength. Also, since there are walls between adjacent cells, the wave does not propagate along the orthogonal direction inside the metasurface. Therefore, all the cells in the bi-anisotropic metasurfaces can be designed individually."

2. The method used to extract the bianisotropic effects differs from that of ref [40], which maintained the boundary condition at the termination of the impedance tube. The reason/justification of using the approach described in this work as opposed to that described in [40] should be provided.

Response: We would like to note that the method used to extract the bianisotropic effects is generally the same as that of ref [40]. The difference is that we employ different boundary conditions. In the approach in ref [40], all the coefficients are obtained by enforcing p_0 and p_5 to

be 0 respectively. Although they maintained the boundary condition at the termination, the incident wave condition is changed. While in our approach, we used the same incident wave and changed the termination. There are two reasons why we use this approach. First, changing terminations using four-microphone method is a standard way of experimentally measuring the acoustic samples, and is widely applied in the commercial acoustic impedance tubes. We tried to stick to this standard to minimize confusion. Second, since we use MATLAB livelink with COMSOL for the optimization, changing boundary conditions in the code is much more convenient and easier to keep track of than changing the direction of the incident wave or the whole structure.

In the revised manuscript, we added a sentence to describe the reason of using the approach described in this work, which reads: “For the ease of implementation, the method we used to retrieve the impedance matrix in COMSOL is the same as the standard 4-microphone method for acoustic experiments with impedance tubes, whose setups are shown in Fig. S2.”

3. Note that Koo et al (ref [39]) used multiple unit cells with membranes to demonstrate bianisotropy, as opposed to a single unit cell as described in the current manuscript. The present manuscript states that [39] only shows the case of a single unit cell and seems to suggest that only [28] proposed use of membranes to achieve acoustic bianisotropy, which is not the case. Pleas correct this in the resubmission.

Response: We would like to thank the reviewer for pointing this out. Here we would like to note that we did not claim to demonstrate the first bianisotropic gradient metasurface, but the first bianisotropic gradient metasurface for perfect manipulation of the wavefront. However, we acknowledge that this part was not well explained. In order to clarify this issue and the confusion, we have extended the discussion in the introduction. The reviewer can see the modifications related with this comment in green color.

4. While the authors do seem to be aware of the relevant previous work on this topic, more clear emphasis should be placed on the enabling experimental demonstrations of refs [39] and [40] and detailed analogies between the electromagnetic and acoustic cases described by [38]. The idea of acoustic bianisotropy was not of clear utility or interest until only very recently, and it was those works that showed formal analogies as well as experimental demonstrations of the behavior.

Response: We understand the concern of the reviewer and we recognize the weaknesses of introductory part in the previous version. The introductory part has been revised to give the reserved recognition the mentioned works. The main modification related to this comment is addressed in the following paragraph:

Interest in bianisotropy in acoustics, also referred to as Willis coupling [37, 38] in elastodynamics, has begun to receive attention [38-40]. Bianisotropy provide two new possibilities in the design of acoustic metasurfaces: one can independently control the reflection and transmission phases, as it was demonstrated by Koo et al. [39], or the difference in the reflection phases, reported by Muhlestein et al. [40]. In addition to these results, a deep analysis of the physics behind this

phenomena and a clear analogy between electromagnetic and acoustic bianisotropy has theoretically studied by Sieck et al. [40]. These results point out that acoustic bianisotropy could bring new directions in the design of more efficient metasurfaces, as it happened in the electromagnetic counterpart.

REVIEWERS' COMMENTS:

Reviewer #1 (Remarks to the Author):

The authors have done an excellent work answering my questions and those of the other referees, and they have also considerably improved the manuscript. I am happy to recommend this work for publication in NC.

Just a last remark about their answer to my comment 4, not really relevant, but perhaps they want to compare the relations for the different coefficients in the framework of PT symmetric systems, just as a matter of notation (perhaps the expression I mentioned to them is not so general). See for instance:

1. Ge, L., Chong, Y. D., & Stone, A. D. (2012). Conservation relations and anisotropic transmission resonances in one-dimensional PT-symmetric photonic heterostructures. *Physical Review A*, 85(2), 023802.
2. Zhu, X., Ramezani, H., Shi, C., Zhu, J., & Zhang, X. (2014). P T-symmetric acoustics. *Physical Review X*, 4(3), 031042.

Especially equation 3 of reference 2.

Reviewer #2 (Remarks to the Author):

The authors responded faithfully to reviewers' comments. The scope of this work became much clearer in the revised version, with proper title.

Meanwhile this reviewer still believe that the novelty of the work does not suffice for 'Nature Communications', considering the practical significance and potential interests in; scattering-free metasurface and practical design of bianisotropic metamaterial, it is suggested to accept this submission for publications.

Minor comments;

1. Reference is missing : at the end of manuscript page 4, "~ based on the generalize refraction law [refs] ~"
2. Figure 5 (c) and (d) have not been revised (old figures are not replaced).

Reviewer #3 (Remarks to the Author):

The revisions provided by the authors have satisfied my concerns. recommend this manuscript for publication in its current form.

Manuscript #: NCOMMS-17-31348A

Title: " Systematic design and experimental demonstration of bianisotropic metasurfaces for scattering-free manipulation of acoustic wavefronts"

Author: Junfei Li, Chen Shen, A. Díaz-Rubio, S. A. Tretyakov, and Steven A. Cummer

Response to Reviewer #1

The authors have done an excellent work answering my questions and those of the other referees, and they have also considerably improved the manuscript. I am happy to recommend this work for publication in NC.

Just a last remark about their answer to my comment 4, not really relevant, but perhaps they want to compare the relations for the different coefficients in the framework of PT symmetric systems, just as a matter of notation (perhaps the expression I mentioned to them is not so general). See for instance:

1. Ge, L., Chong, Y. D., & Stone, A. D. (2012). Conservation relations and anisotropic transmission resonances in one-dimensional PT-symmetric photonic heterostructures. *Physical Review A*, 85(2), 023802.

2. Zhu, X., Ramezani, H., Shi, C., Zhu, J., & Zhang, X. (2014). P T-symmetric acoustics. *Physical Review X*, 4(3), 031042.

Especially equation 3 of reference 2.

The paper is interesting and the results consistent with the theory, although there are some points that have to be clarified before the paper be accepted for NC.

Response: The referee raised an interesting view point of our work. We have added some discussion about PT symmetric systems and the manuscript is revised as follows: "It is interesting to note that there the asymmetric matching conditions required for perfect anomalous transmission can also be fulfilled with the use of PT-symmetric structures [45]. In electromagnetics, this possibility was previously noted [30, 46]. However, realizations of this PT approach require the use of active elements."

In the revised manuscript, we now further discuss the relation between the reflection case and transmission case, which reads: "It is worth noting here that the reflection coefficient has twice the tangential wave number of the transmission coefficient, indicating that energy exchange along the metasurface is carried out by surface waves on the reflection side. This is related to the idea of

controlling reflection by launching an auxiliary surface wave to achieve scattering-free wave manipulation [32]”

Response to Reviewer #2

The authors responded faithfully to reviewers' comments. The scope of this work became much clearer in the revised version, with proper title.

Meanwhile this reviewer still believe that the novelty of the work does not suffice for 'Nature Communications', considering the practical significance and potential interests in; scattering-free metasurface and practical design of bianisotropic metamaterial, it is suggested to accept this submission for publications.

Minor comments;

1. Reference is missing: at the end of manuscript page 4, “~ based on the generalize refraction law [refs] ~”

Response: We apologize for the typo. The corresponding reference is added to the revised manuscript.

2. Figure 5 (c) and (d) have not been revised (old figures are not replaced).

Response: The figures have been updated in the revised manuscript with the new version. In addition, we also added the color bar and unit in the new figure.

Response to Reviewer #3

The revisions provided by the authors have satisfied my concerns. recommend this manuscript for publication in its current form.

Response: We would like to thank the referee for the positive remarks on our work.